# Convcast: An embedded convolutional LSTM based architecture for precipitation nowcasting using satellite data

Ashutosh Kumar[1,2]*, Tanvir Islam[1], Yoshihide Sekimoto[2], Chris Mattmann[1], Brian Wilson[1]

**1** Jet Propulsion Laboratory, California Institute of Technology, Pasadena, CA, United States of America, **2** Institute of Industrial Science, The University of Tokyo, Tokyo, Japan

\* ashutosh@iis.u-tokyo.ac.jp

**Data Availability Statement:** Dataset used for precipitation nowcasting in this research is publicly available at https://pmm.nasa.gov/data-access/downloads/gpm.

## Abstract

Nowcasting of precipitation is a difficult spatiotemporal task because of the non-uniform characterization of meteorological structures over time. Recently, convolutional LSTM has been shown to be successful in solving various complex spatiotemporal based problems. In this research, we propose a novel precipitation nowcasting architecture 'Convcast' to predict various short-term precipitation events using satellite data. We train Convcast with ten consecutive NASA's IMERG precipitation data sets each at intervals of 30 minutes. We use the trained neural network model to predict the eleventh precipitation data of the corresponding ten precipitation sequence. Subsequently, the predicted precipitation data are used iteratively for precipitation nowcasting of up to 150 minutes lead time. Convcast achieves an overall accuracy of 0.93 with an RMSE of 0.805 mm/h for 30 minutes lead time, and an overall accuracy of 0.87 with an RMSE of 1.389 mm/h for 150 minutes lead time. Experiments on the test dataset demonstrate that Convcast consistently outperforms other state-of-the-art optical flow based nowcasting algorithms. Results from this research can be used for nowcasting of weather events from satellite data as well as for future on-board processing of precipitation data.

## Introduction

Precipitation nowcasting refers to the prediction of rainfall in a local region over a short period of time generally up to six hours [1]. Short-term prediction of weather events is important for public safety from high-impact meteorological events such as flash floods, tropical cyclones, thunderstorms, lightning, high-speed wind, etc. which can affect large population or areas of significant economic investment. Precipitation nowcasting is also useful for weather forecasts and guidance in aviation, marine safety, ground traffic control, and construction industries. Several outdoor activities such as trekking, rafting, fishing also depend on short-term forecasts of weather events. In recent years, rapid climate change has also led to catastrophic

**Funding:** This research was funded by Jet Propulsion Laboratory, California Institute of Technology (https://jpl.nasa.gov) under JPL DSWG Pilot Project. The funders had no role in study design, data collection and analysis, decision to publish, or preparation of the manuscript.

**Competing interests:** The authors have declared that no competing interests exist.

meteorological events such as flash floods in various parts of the world because of the unusual precipitation [2]. Accurate and reliable nowcasting of weather data is thus important.

Nowcasting is one of the most challenging problems in weather forecasting because of the non-uniform and flawed characterization of the meteorological structures over time. Traditional methods for forecasting based on Numerical Weather Prediction (NWP) are not suitable for short-term predictions because they are highly computationally expensive, sensitive to noise and depends a lot on initial conditions of the event [3]. They cause a delay in short-term predictions because of data assimilation and simulation steps required in NWP models which make the forecast irrelevant by the time it is made. Traditionally, nowcasting is done using Radar Echo Extrapolation (REE) which are reflected electromagnetic waves from the hydrometeors in the atmosphere. Radar echoes are very detailed and can give the intensity, speed, shape, and direction of movement of storms continuously. The approach of extrapolating the radar echoes tends to outperform NWP based models [4] and is currently the state-of-the-art method in precipitation nowcasting [5–8]. The method of extrapolation of radar echoes gives the movement and change in intensity of hydrometeors from radar images. Computer vision based technique optical flow has been widely used to extrapolate radar maps in nowcasting [9–11]. However, there are some limitations of nowcasting of precipitation using radar echoes. Radar echoes are best suitable for a range between 5 kms to 200 kms on earth. Beyond 200 kms, they can detect rainfall that are at a higher altitude and does not reflect the real scenario on the ground. Sometimes within optimal range, there is a possibility of detection of *virga* which is rainfall evaporating before hitting the ground, and reflection from objects that are not rainfall such as buildings, drones, airplanes, birds [12]. Radar cannot measure the rainfall directly above the observatory. Another major limitation of radar-based precipitation nowcasting is that radar observatories are not present in many developing nations and are also not available over deep oceans. Satellite-based precipitation data for nowcasting can be beneficial in those cases as they cover the entire globe.

Several studies have used satellite-based data for precipitation nowcasting [13–16]. In [13], the authors used picture fuzzy clustering method for weather nowcasting from satellite image sequences. Rivolta et al. [14] used a two-step approach by first projecting the infrared radiance measured from satellite ahead in time and then use the projected radiance to nowcast further using artificial neural network. Otsuka et al. [15] used data assimilation technique for nowcasting from satellite images. Liu et al. [16] used computer vision technique optical flow for precipitation nowcasting using satellite data. Methodologies used in [13, 15] are computationally expensive and require several processing steps to obtain the nowcasting results. Rivolta et al. [14] used simple feed-forward neural network without any consideration for spatiotemporal aspects which leads to large error in nowcasting results. Optical flow based techniques in nowcasting have limited success results because the tracking of pixels and extrapolation of values during prediction are considered as two separate processes [17, 18].

Recently, deep learning techniques have been shown to be very successful in solving several real-world problems such as image classification, object detection, image captioning, text analysis in computer vision and natural language processing [19]. Deep neural networks have the ability to capture intricate structures from the dataset to learn a function using the backpropagation algorithm that can map the input to the output. Recurrent neural networks such as Long Short Term Memory (LSTM) [20], and Gated Recurrent Units (GRU) [21] are widely used for analyzing time series and sequential data. However, LSTMs and GRUs can not work with problems which have spatiotemporal aspects such as sequential radar maps for nowcasting [17]. Some studies tried to use Convolutional Neural Network (CNN) for extracting local spatial information and LSTM for long distance dependency [22]. In those studies, CNNs and LSTMs are treated independently as they considered output from CNN as input to the LSTM.

Convolutional LSTM (ConvLSTM) [17] has convolutional structures embedded within the LSTM cell. Such network architecture has been shown to be very successful in extracting spatiotemporal features required for precipitation nowcasting using radar data [17, 23, 24]. A comprehensive study of ConvLSTM based architectures and its comparison with state-of-the-art methods in precipitation nowcasting using satellite data is required.

Building a deep neural network architecture for a new problem requires a lot of trial and error effort because of several hyperparameters embedded in the network. Traditionally, hyperparameters optimization has been done by humans when only a few trials are possible. Automated Machine Learning (AutoML) tries to solve the manual effort by automatically optimizing the architecture by tuning its hyperparameters on a given dataset to achieve optimal performance [25]. Recently, there has been an increase in computing capacity because of the parallelization on GPU processors which makes it possible to run several trials in a small time duration. Algorithmic-based approaches for hyperparameter optimization has been shown to be quite successful in various problems [26]. However, to the best of our knowledge, we could not find previous studies which uses AutoML for precipitation nowcasting problems.

In this research, we develop a novel architecture Convcast for precipitation nowcasting using the spaceborne Integrated Multi-satellitE Retrievals for GPM (IMERG) dataset [27]. We (i) use Convcast to predict the eleventh timestamp precipitation from a series of ten consecutive precipitation data at an interval of 30 minutes, (ii) use the predicted precipitations sequentially to further nowcast 60 minutes, 90 minutes, 120 and 150 minutes lead time precipitations, (iii) investigate the role of different hyperparameters in the Convcast network, and (iv) assess the quality of the nowcasted precipitation with the state-of-the-art optical flow based baseline methods on the test dataset consisting of several meteorological events.

## Methodology

### IMERG dataset

IMERG is the unified algorithm that provides multi-satellite precipitation data. The precipitation data is obtained from passive microwave sensors of the precipitation measuring satellite comprising the Global Precipitation Measurement (GPM) constellation [27]. The IMERG dataset is available in temporal resolutions of 30 minutes, 3 hours, 1 day, 7 days, and 30 days. All IMERG dataset has a spatial resolution of 0.1˚. Since our goal is short-term forecasting of precipitation, we use the IMERG dataset with a temporal resolution of 30 minutes. The dataset with a temporal resolution of 30 minutes are available since March 2014. The IMERG dataset has a width of 3600 gridded values and a height of 1800 gridded values which cover the latitude from -89.95 to 89.95 and longitude from -179.95 to 179.95 on the earth.

IMERG dataset with a temporal resolution of 30 minutes is available in HDF5, GeoTIFF, NetCDF, ASCII, PNG, KMZ, OpenDAP, GrADS and THREDDS data formats. For our research, we use the HDF5 format IMERG dataset [28] for all subsequent analysis. We use only the '*precipitatonCal*' field from the HDF5 dataset which is multi-satellite precipitation data with gauge calibration and has a unit of mm/hour.

### Nowcasting problem and training data

In a precipitation nowcasting problem using satellite data, the spatial region is represented by M x N grid with Z measurement values varying over time. At any time t, the observation is a tensor X where $X \in R^{M \times N \times Z}$ where $R$ is the observed feature (precipitation). If the observation is recorded periodically, we get a sequence of observed features $X^{<1>}, X^{<2>}, X^{<3>}, \ldots, X^{<t>}$. The nowcasting problem is then to predict the next sequence $X^{<t+1>}$ given the previous observations. In this research, we choose a square grid (M = N = 150) by cropping squares of

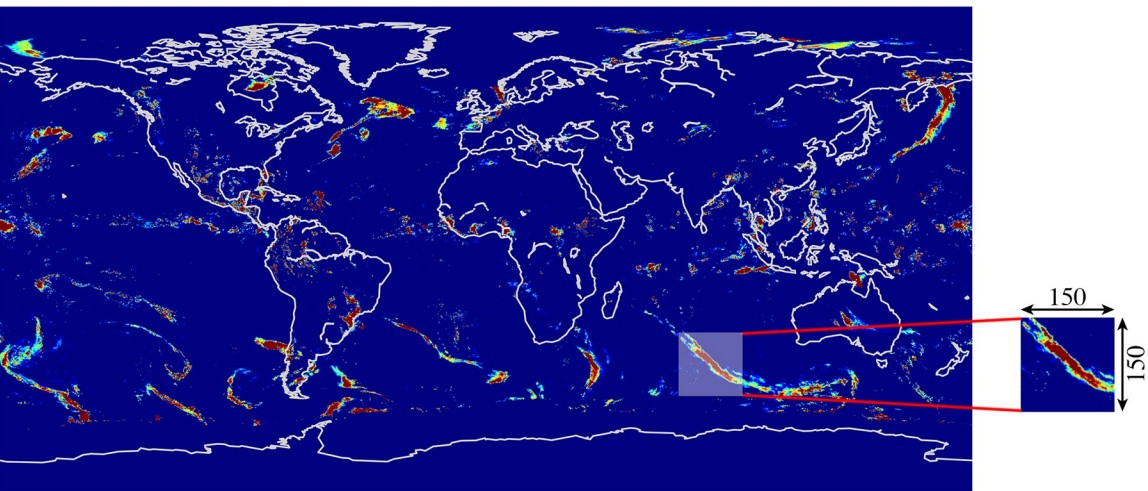

**Fig 1. 150 x 150 crops from '*precipitatonCal*' variable of the IMERG HDF5 file.**

resolution 150 x 150 from the IMERG dataset of size 3600 x 1800 with '*precipitatonCal*' (Z = 1) varying over time as shown in Fig 1.

In our study, we would like to predict the sequence $X^{<11>}$ from previous ten observations at an interval of 30 minutes. For each input precipitation data, we use the subsequent precipitation data as the output precipitation in the training set. For example, at the fifth timestamp $X^{<5>}$, we pass the following 30-minute i.e. the sixth precipitation data $X^{<6>}$ as the output label. Therefore, we prepare ten consecutive precipitation data each at an interval of 30 minutes from the IMERG dataset as shown in Fig 2. We prepare 1,276 examples in the training set, 319 examples in the validation set, and 242 examples for the test set. All three sets in training, validation, and testing have diverse sets of precipitation examples such as hurricanes, storms, tropical depression, etc.

## Development of the Convcast architecture using AutoML

Convcast consists of several layers of ConvLSTM cells for precipitation nowcasting. Unlike normal LSTM cells, ConvLSTM has convolutional structures in input-to-state and state-to-state transitions for modeling spatiotemporal relationships. In a ConvLSTM cell, the inputs $X^{<1>}, X^{<2>}, \ldots, X^{<t>}$, cell outputs $C^{<1>}, C^{<2>}, \ldots, C^{<t>}$, hidden states $H^{<1>}, H^{<2>}, \ldots, H^{<t>}$, input gate $\Gamma_i$, forget gate $\Gamma_f$, and output gate $\Gamma_o$ are three-dimensional tensors. The key

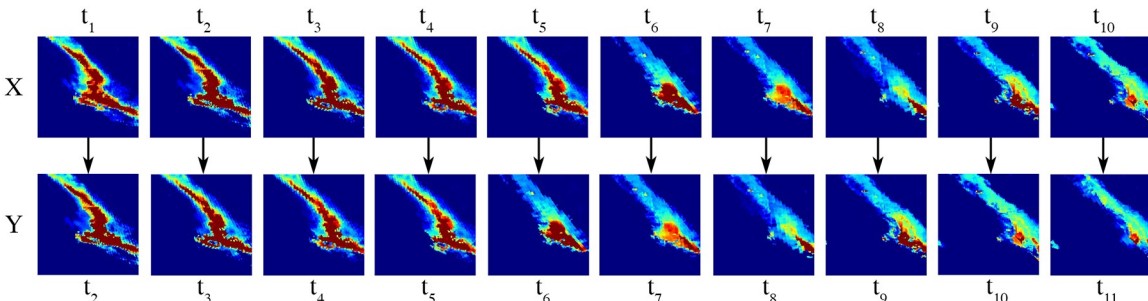

**Fig 2. Each timestamp $X^{<t>}$ gets the label of the next consecutive timestamp $X^{<t+1>}$ for prediction.**

equations governing the ConvLSTM cells are shown in the set of Eq 1, where ∘ denotes the element-wise multiplication (Hadamard product), * denotes the convolution operation, $\sigma$ represents the sigmoid function, tanh represents the hyperbolic tangent function and **W**, **b** are the parameters of the neural network.

$$
\left.
\begin{aligned}
\Gamma_i &= \sigma(W_{xi} * X^{<t>} + W_{hi} * H^{<t-1>} + W_{ci} \circ C^{<t-1>} + b_i) \\
\Gamma_f &= \sigma(W_{xf} * X^{<t>} + W_{hf} * H^{<t-1>} + W_{cf} \circ C^{<t-1>} + b_f) \\
C^{<t>} &= \Gamma_f \circ C^{<t-1>} + \Gamma_i \circ \tanh(W_{xc} * X^{<t>} + W_{hc} * H^{<t-1>} + b_c) \\
\Gamma_o &= \sigma(W_{xo} * X^{<t>} + W_{ho} * H^{<t-1>} + W_{co} \circ C^{<t>} + b_o) \\
H^{<t>} &= \Gamma_o \circ \tanh(C^{<t>})
\end{aligned}
\right\}
\tag{1}
$$

We develop Convcast by stacking three ConvLSTM layers for spatial and temporal learning feature learning which followed by a 3D convolutional layer for the next 30 minutes precipitation prediction as shown in Fig 3. In the last layer of the Convcast architecture, we use ReLU as the activation layer. This is because precipitation nowcasting is a regression problem where the output of the Convcast is a precipitation value. Since precipitation cannot take negative values, we choose ReLU to turn any negative activations into zeros (i.e. no rain). For selecting the best possible hyperparameters in Convcast architecture, we do a random search [26] for 200 iterations on the set of hyperparameters using the training dataset as shown in Table 1 and measure the mean squared error loss on the validation dataset. The maximum number of epochs considered during hyperparameter optimization is 20 as validation loss converges around 15 epochs as shown in Fig 4. For hyperparameter optimization, we use an open-source AutoML toolkit Neural Network Intelligence (NNI) [29]. It is important to note here that during our experiment we find that a learning rate of 0.001 diverges the training loss regardless of the other hyperparameters in the network. Subsequently, we remove this hyperparameter from our search space.

For running our experiments we use a computer system with the following specifications: NVIDIA Tesla V100 with 16 GB GPU memory, High Frequency Intel Xenon E5-2686 v4 (Broadwell) 2.7 GHz processor with 64GB RAM. The hyperparameters used in Convcast after optimization in NNI framework are shown in Table 2.

## Baseline methods

As a state-of-the-art baselines, we employ four optical flow based models as discussed in [18]. Optical flow based nowcasting algorithms have two steps which includes tracking and extrapolation of features. The four models [18] considered as baselines in this research are Sparse Single Delta (SparseSD), Sparse, Dense, and Dense Rotation (DenseROT). Unlike our Convcast model, these methods are unsupervised algorithms. It is important to note here that only Sparse model can use last $t - 10$ precipitation data for the next $t + 1$ consecutive prediction. All other models use only the last $t - 2$ precipitation values for the $t + 1$ prediction.

The tracking and extrapolation algorithm used in the four models are listed in Table 3. Both SparseSD and Sparse use Lucas-Kanade optical flow algorithm [30] to track features calculated using Shi-Tomasi corner detector [31]. In case of SparseSD, a constant displacement is calculated for each tracked feature which is then propagated linearly for predicting location of features for each lead time $n$. An affine transformation matrix is then calculated for each lead time $n$ based on the location of identified features in the previous step. The precipitation at time $t + n$ is calculated by multiplying affine transformation matrix with the precipitation at time $t$. Sparse model differs from SparseSD that instead of using a constant displacement for

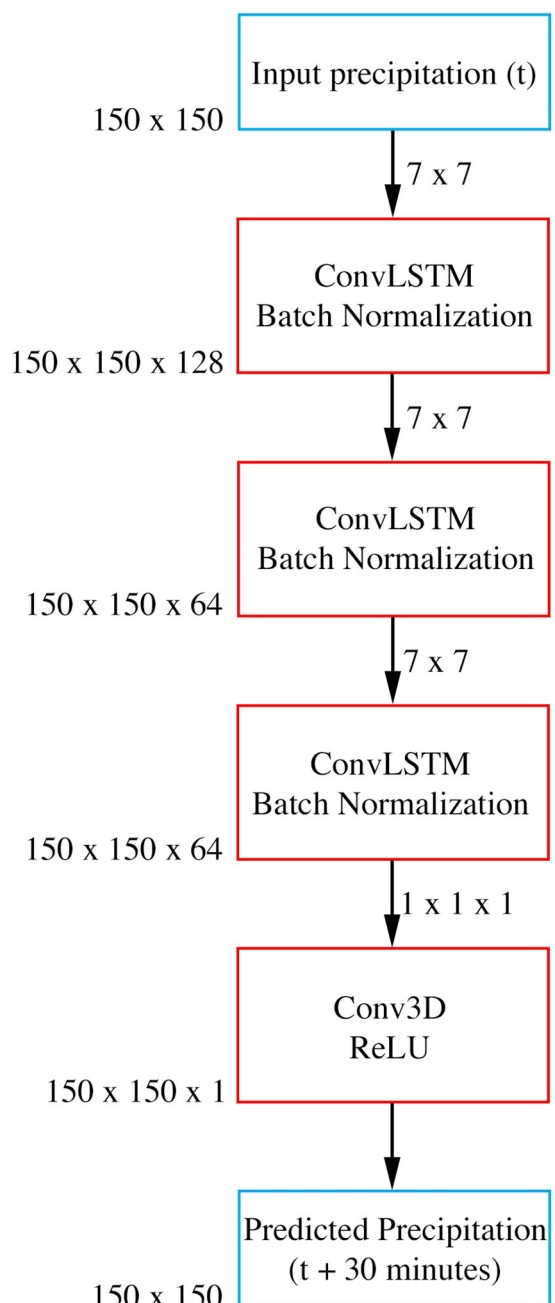

**Fig 3. Convcast architecture for precipitation nowcasting using the IMERG dataset.**

**Table 1. Hyperparameter space for random search.**

| Learning Rate | Filters | Kernel Size | Recurrent Regularizer | Kernel Regularizer | Recurrent Dropout |
|:---:|:---:|:---:|:---:|:---:|:---:|
| 0.001 | 16 | 2x2 | 0.0 | 0.0 | 0.0 |
| 0.0001 | 32 | 3x3 | 0.2 | 0.2 | 0.2 |
| - | 64 | 5x5 | 0.3 | 0.3 | 0.3 |
| - | 128 | 7x7 | - | - | - |

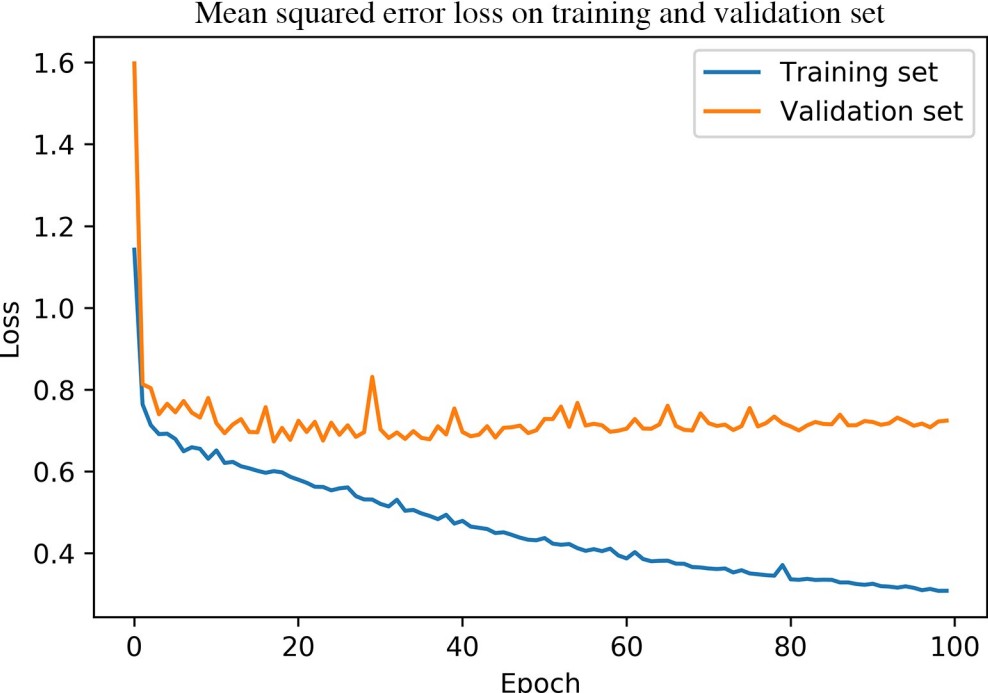

**Fig 4. Convergence of validation loss after 15 epochs for a ConvLSTM based model.**

each feature, Sparse model uses a linear regression model for every tracked features through time $t - 10$ to $t$.

Dense and DenseROT models use Dense Inverse Search optical flow algorithm [32] for calculating displacement field using the previous $t - 1$ timestamps. The pixel values are then extrapolated according to the displacement field using a backward constant-vector [10] and

**Table 2. Hyperparameters in Convcast architecture.**

| Hyperparameters | Value |
|---|---|
| Learning rate | 0.0001 |
| Batch Normalization | True |
| Batch size | 2 |
| Loss function | MSE |
| Activation function | tanh (ConvLSTM), ReLU (Conv3D) |
| Optimizer | Adam |
| Hidden layers | 4 |
| Input data size | 150 x 150 |
| Number of filters (Input-to-state) | 128 |
| Number of filters (State-to-state) | 64 |
| Kernel size (Input-to-state) | 7 x 7 |
| Kernel size (State-to-state) | 7 x 7 |
| Dropout | False |
| Regularizer | False |
| Feature scaling | True [0, 1] |

**Table 3. Overview of the four baseline methods based on optical-flow algorithm.**

| Models | Tracking algorithm | Extrapolation |
|---|---|---|
| SparseSD | Shi-Tomasi corner detector with Lucas-Kanade | Constant delta-change and affine transformation |
| Sparse | Shi-Tomasi corner detector with Lucas-Kanade | Linear regression and affine transformation |
| DenseROT | Dense Inverse Search | Semi-Lagrangian advection |
| Dense | Dense Inverse Search | Constant-vector advection |

semi-Lagrangian scheme [8] in case of Dense and DenseROT, respectively. It should be noted that a constant-vector approach does not allow for the representation of roational motion while a semi-Lagrangian consider rotational motion with the assumption of motion field to be persistent.

Apart from the above optical flow baseline methods, we also use a simple LSTM layer network for nowcasting. In this case, we flatten the input IMERG dataset of shape 150 x 150 to 22,500 cells and build a two layered LSTM network with 2048 units in the first layer and 1024 units in the second layer. We then pass it through a fully connected layer with 22,500 units to predict the next 30-minute precipitation.

## Evaluation on the test dataset

We nowcast the precipitation up to 150 minutes from the tenth precipitation data in the sequence, as shown in Table 4. We assess the quality of the nowcasted precipitation up to 150 minutes using two methods. The first method is based on the dichotomous approach (binary classification), where we convert the predicted and observed matrix to a binary matrix consisting of zeros and ones. If a cell in the matrix has a positive value greater than zero (detected rain), we assign a value one, otherwise zero (no rain). We compare each pixel value of the observed and predicted binary matrices to form the contingency matrix, as shown in Table 5 and calculate accuracy indices.

The second method is based on the forecast of continuous variables, and it measures the difference between the magnitude of observed and forecasted values. In this case, we form a one-dimension array of size N (N = 5,445,000) from the 242 predicted and the observed precipitation matrix of size 150 x 150 and compute the accuracy indices directly from the observed and predicted values.

**Accuracy indices based on dichotomous (binary classification) method.** From the variables in the contingency matrix as shown in Table 5, we calculate a total of nine accuracy scores for evaluation of the model.

**Table 4. Dataset sequence for nowcasting up to 150 minutes.**

| Timestamp(Nowcasted) | Time difference from $t_{10}$ | Sequences used (Convcast, Sparse, LSTM) | | Sequences used (DenseROT, Dense, SparseSD) | |
|---|---|---|---|---|---|
| | | Observed | Predicted | Observed | Predicted |
| $t_{11}$ | 30 minutes | $t_1$ to $t_{10}$ | - | $t_9, t_{10}$ | - |
| $t_{12}$ | 60 minutes | $t_2$ to $t_{10}$ | $t_{11}$ | $t_{10}$ | $t_{11}$ |
| $t_{13}$ | 90 minutes | $t_3$ to $t_{10}$ | $t_{11}, t_{12}$ | - | $t_{11}, t_{12}$ |
| $t_{14}$ | 120 minutes | $t_4$ to $t_{10}$ | $t_{11}, t_{12}, t_{13}$ | - | $t_{12}, t_{13}$ |
| $t_{15}$ | 150 minutes | $t_5$ to $t_{10}$ | $t_{11}, t_{12}, t_{13}, t_{14}$ | - | $t_{13}, t_{14}$ |

**Table 5. Contingency matrix for the dichotomous forecast.**

| | | Observed (IMERG) | | Total |
|---|---|---|---|---|
| | | 1 | 0 | |
| Predicted | 1 | Hits | False Alarms | Predicted 'Yes' |
| | 0 | Misses | True Negatives | Predicted 'No' |
| Total | | Observed 'Yes' | Observed 'No' | Total (N) |

Accuracy is the fraction of correct forecasts by the model. This metric is heavily affected by the dominating class (e.g. no precipitation).

$$Accuracy = \frac{Hits + True\ Negatives}{Total} \tag{2}$$

Bias measures the ratio of the frequency of forecasted events to the observed events. A value greater greater than one indicates over forecast and a value less than one indicates under forecast.

$$Bias = \frac{Hits + False\ Alarms}{Hits + Misses} \tag{3}$$

Equitable Threat Score (ETS) measures the forecast skill accounting for the hits due to random chance. It is also used to compare forecasts in different regimes.

$$ETS = \frac{Hits - Hits_{Random}}{Hits + Misses + False\ Alarms - Hits_{Random}}\ ,\ where$$

$$Hits_{Random} = \frac{(Hits + Misses)(Hits + False\ Alarms)}{Total} \tag{4}$$

False Alarm Ratio (FAR) is the fraction of predicted rainfall events that actually did not occur.

$$FAR = \frac{False\ Alarms}{Hits + False\ Alarms} \tag{5}$$

Heidke Skill Score (HSS) measures the fraction of correct forecasts after removing those forecasts that may be due to random chance.

$$HSS = \frac{(Hits + True\ Negatives) - (Random\ Correct)}{N - (Random\ Correct)}, where$$

$$Random\ Correct = \frac{1}{N}[(Hits + False\ Alarms)(Hits + Misses)$$

$$+ (True\ Negatives\ + False\ Alarms)(True\ Negatives\ + Misses)] \tag{6}$$

Odds Ratio Skill Score (ORSS) is also known as Yule's Q and it measures the improvement of forecast over random chance.

$$ORSS = \frac{True\ Negatives\ \times\ Hits\ -\ False\ Alarms\ \times\ Misses}{True\ Negatives\ \times\ Hits\ +\ False\ Alarms\ \times\ Misses} \tag{7}$$

Probability of False Detection (POFD) is also known as false alarm rate and it measures the precipitation events that were not observed but were incorrectly forecasted as precipitation

events.

$$POFD = \frac{False\ \ Alarms}{True\ \ Negatives + \ \ False\ \ Alarms} \tag{8}$$

Success Ratio measures the fraction of the forecasted precipitation events that were correctly observed.

$$Success\ \ Ratio = \frac{Hits}{Hits\ \ + \ \ False\ \ Alarms} \tag{9}$$

Threat Score is also known as Critical Success Index (CSI) and it represents how well the forecasted 'Yes' precipitation events correspond to the observed 'Yes' precipitation events.

$$Threat\ \ Score/CSI = \frac{Hits}{Hits\ \ + \ \ False\ \ Alarms + \ \ Misses} \tag{10}$$

**Accuracy indices based on continuous variables.** For calculation of the accuracy indices based on continuous variables, we compare each value ($P_i$) in the forecasted matrix with corresponding values ($O_i$) in the observed IMERG matrix and calculate three accuracy scores for the model performance. Root Mean Square Error (RMSE) measures the average magnitude of forecast errors. The effect of each error on RMSE is proportional to the size of the squared error. It should be noted that it is influenced heavily by large errors than smaller errors.

$$RMSE = \sqrt{\frac{1}{N}\sum_{i=1}^{N}(P_i - O_i)^2} \tag{11}$$

Multiplicative Bias compares the average value of forecast with the average value of observed precipitation.

$$Multiplicative\ \ Bias = \frac{\sum_{i=1}^{N}P_i}{\sum_{i=1}^{N}O_i} \tag{12}$$

Correlation Coefficient indicates the correspondence of the predicted values to the observed values and shows the phase relationship between them.

$$Correlation\ \ Coefficient = \frac{\sum_{i=1}^{N}(P_i - \bar{P})(O_i - \bar{O})}{\sqrt{\sum_{i=1}^{N}(P_i - \bar{P})^2}\sqrt{\sum_{i=1}^{N}(O_i - \bar{O})^2}} \tag{13}$$

## Results

We calculate the MSE loss on the validation set for the simple LSTM model, a ConvLSTM model with a kernel size of 2 x 2 and 5 x 5, and the Convcast. We present the validation loss of the four models in Table 6. From the table, it is evident that simple LSTM has a large error on the validation dataset indicating that such model cannot capture spatial information because of the flattening of the input precipitation data. We find that even a ConvLSTM architecture with a small kernel size of 2 x 2 outperforms the simple LSTM model by 47.61%. The loss of ConvLSTM based model decreases with the increase in the kernel size (5 x 5) and the number of filters (64). Convcast achieves the minimum loss on the validation set. We do not further present simple LSTM results on the test dataset because it achieves large errors similar to the validation set and we only choose Convcast to evaluate on the test dataset as it achieves the

**Table 6. Comparison of validation loss of Convcast with other LSTM models.**

| Models | Validation loss |
| --- | --- |
| LSTM (-2048-1024-[22, 500]) | 1.569 |
| ConvLSTM (2x2-32-2x2-32-2x2-32-1x1x1) | 0.822 |
| ConvLSTM (5x5-64-5x5-64-5x5-64-1x1x1) | 0.626 |
| **Convcast** | **0.6049** |

'-2048' and '-1024' represents the number of units in LSTM cell '-[22, 500]' is the number of units in the fully connected layer. '2x2' and '5x5' represent input-to-state kernel size and '-2x2' and '-5x5' represent state-to-state kernel size. '-32' and '-64' are the number of filters in the hidden states of the ConvLSTM layers. '-1x1x1' refers to the 3D convolution.

minimum loss on the validation dataset. It should also be noted that the optical flow based baseline methods do not require supervised learning and thus we only show their results on the test dataset. We calculate the accuracy scores on the test dataset using Convcast and four other baseline methods. The accuracy scores based on dichotomous and continuous variables are shown in Figs 5 and 6, respectively. From Fig 5, we find that Convcast has better results compared to other baseline methods. Only in the case of Odds Ratio Skill Score for 30 minutes

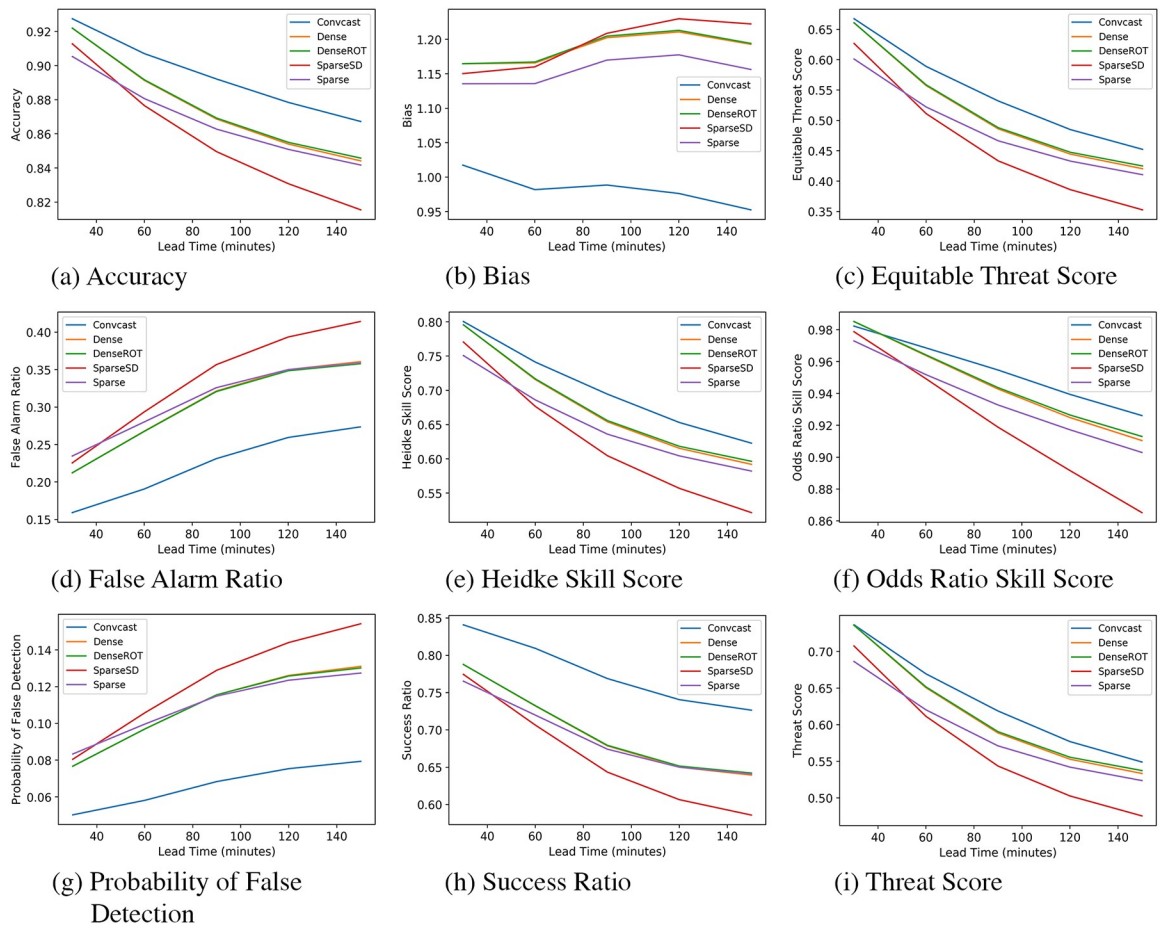

**Fig 5. Variation of dichotomous accuracy scores with time.**

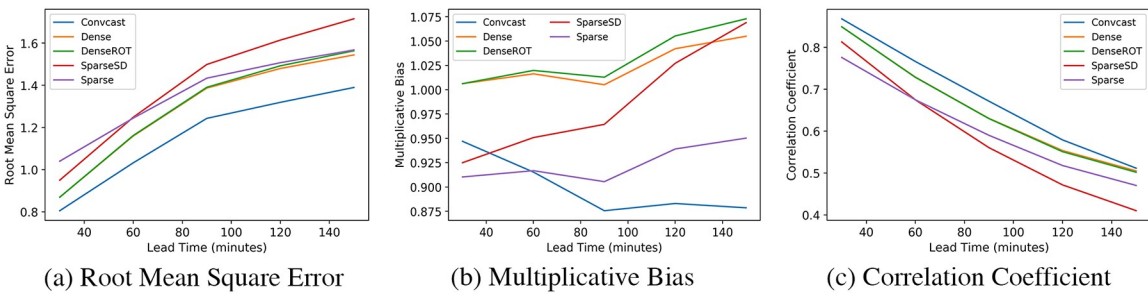

(a) Root Mean Square Error (b) Multiplicative Bias (c) Correlation Coefficient

**Fig 6. Variation of accuracy scores based on continuous variables with time.**

lead time, Dense groups have more skills compared to Convcast. Similarly, from Fig 6 we find that Convcast outperforms other baseline methods for 30 minutes lead time nowcasting.

In the case of dichotomous forecast results, we find that all accuracy scores except False Alarm Ratio and Probability of False Detection decreases with time as shown in Table 7. It should be noted that similar to the experiments in [18], there is negligible difference between Dense and DenseROT models and they outperform Sparse group models. DenseROT achieves slightly better results compared to Dense model because of rotation motion consideration. Thus, we only compare DenseROT with Convcast for percentage change in forecast metric in Table 7.

From Table 7, we find Convcast dominates DenseROT for less decrease in the accuracy scores or more increase in error metrics as we forecast further in time. The decrease in accuracy scores is expected with further forecasting in time [13, 14, 17]. It has been further illustrated with an example in the test dataset of a storm nowcasted from t + 30 minutes to t + 150 minutes using Convcast as shown in Fig 7. It should be noted that the precipitation values in Fig 7 have been scaled between zero and five for better visualization. It is evident from Fig 7 that as we forecast further in time, the accuracy of the model decreases. The model predicts precipitation values well initially up to t + 90 minutes, but beyond t + 90 minutes, the model struggles with the prediction of the magnitude of the precipitation value. It just tends to average the values from previous precipitation which is evident from the blurry images. Interestingly, in all cases, the model preserves the direction and the speed of the storm.

**Table 7. Percentage increase ↑ and decrease ↓ in the forecast metric scores.**

| Metric | Convcast ($t_{11}$ to $t_{15}$) | DenseROT ($t_{11}$ to $t_{15}$) |
|---|---|---|
| Accuracy | **6.490 ↓** | 8.258 ↓ |
| Bias | 6.377 ↓ | **2.529 ↑** |
| Equitable Threat Score | **32.213 ↓** | 35.669 ↓ |
| False Alarm Ratio | 71.885 ↑ | **68.592 ↑** |
| Heidke Skill Score | **22.180 ↓** | 25.031 ↓ |
| Odds Ratio Skill Score | **5.720 ↓** | 7.327 ↓ |
| Probability of False Detection | **57.987 ↑** | 69.704 ↑ |
| Success Ratio | **13.601 ↓** | 18.477 ↓ |
| Threat Score | **25.440 ↓** | 26.961 ↓ |
| Root Mean Square Error | **72.567 ↑** | 79.833 ↑ |
| Multiplicative Bias | 7.217 ↓ | **6.625 ↑** |
| Correlation Coefficient | 41.058 ↓ | **40.893 ↓** |

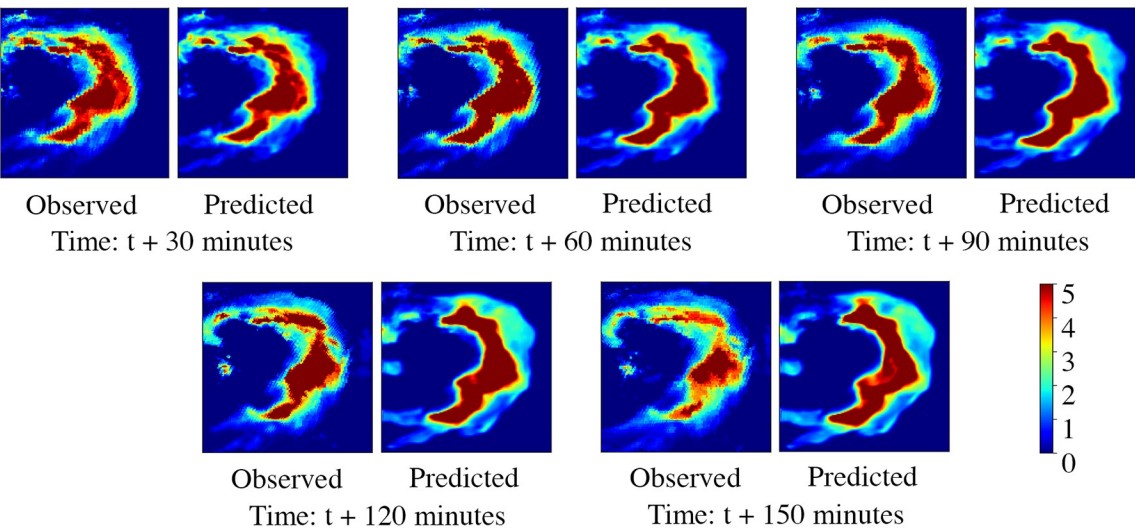

**Fig 7. Nowcasting of a storm up to 150 minutes lead time using Convcast.**

## Discussion

Results of comparison among simple LSTM and ConvLSTM models suggest that simple LSTM models are not suitable for spatiotemporal problems and even a kernel with a small receptive field in ConvLSTM outperforms simple LSTM models. This is expected as for simple LSTM models we flatten the input (150 x 150) precipitation data into a vector of length 22,500 which breaks all the spatial correlations. However, nowcasting requires preservation of spatial features as for a given event in a local region, the motion of cloud is consistent. ConvLSTM,

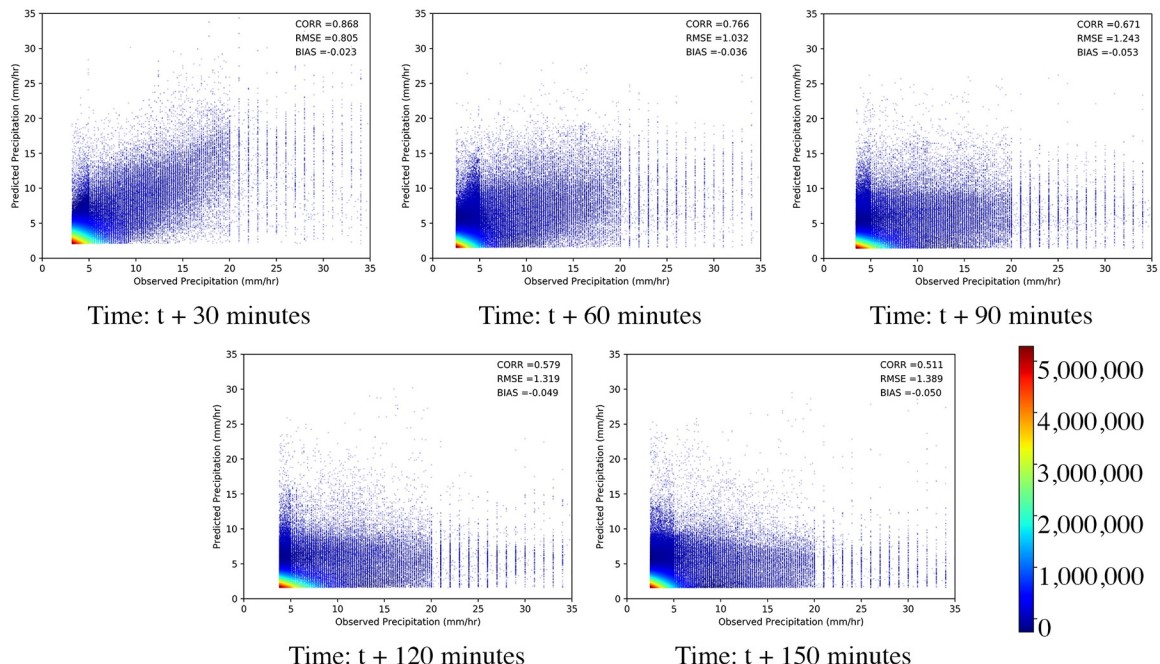

**Fig 8. Scatter density plots of magnitude of observed precipitation and predicted precipitation with lead time.**

because of the embedded convolutional structures as shown in Eq 1, can preserve the spatial information even with a small kernel of size 2 x 2. The accuracy of the ConvLSTM model increases with increasing kernel size and the number of filters because a large kernel has more receptive field and can capture more spatial information and larger number of filters can have more number of feature maps. Since the ConvLSTM based architectures as shown in Table 6 have same backbone in terms of number of layers and activation functions as Convcast, the least validation loss obtained by Convcast is due to the tuned hyperparameters obtained from the AutoML experiment.

In general, the accuracy of the model tends to improve with an increasing kernel size but that also increases the number of parameters of the model making it more complex and causes the training and testing time to increase. From the hyperparameter tuning experiment, we find that the accuracy of the model increases with increasing kernel size but we do not measure a significant validation score difference between kernels of size 7 x 7 and 9 x 9 and we choose kernel size of 7 x 7 in Convcast because of lesser number of parameters compared to a kernel of size 9 x 9.

We find that Convcast outperforms optical flow based baseline methods both in dichotomous and continuous metric scores. Similar to experiments in [18] on radar dataset, we find that SparseSD model outperforms Sparse model for immediate t + 30 minutes lead time precipitation and it is outperformed by Sparse model for longer lead time precipitation. Better performance of Sparse model for longer lead time nowcasting could be because of the larger previous input precipitation information in the Sparse model which generalizes well for a longer lead time precipitation. Differences in Sparse groups (SparseSD and Sparse) and Dense groups (DenseROT and Dense), could be because of the local features detected by Shi-Tomasi corner detector does not represent the overall precipitation event as mentioned in [18]. In our studies, we find that DenseROT performs similar to Dense model despite simpler extrapolation technique in Dense model as compared to DenseROT as shown in Table 3. As mentioned in [8] that despite theoretical superiority of Semi-Lagrangian advection scheme compared to Constant-vector advection, they have similar performance because of the same interpolation technique in them.

Differences between optical flow based approaches and Convcast is expected as Convcast has been trained end-to-end using a large number of precipitation events that comprise several precipitation. Boundary conditions, for example, sudden appearance of clouds are well handled by Convcast because of presence of such examples in the training set. Such complex situations, however, cannot be handled by optical flow based methods. Further, because Convcast is trained end-to-end, several intricate structures present in the precipitation data can be captured by the non-linear and convolutional structures [17].

From the nowcasting results on the test dataset, we find that the accuracy of nowcasting decreases as we forecast further in time. In the case of dichotomous based forecasting, we find that all accuracy scores decrease with lead forecast time. The HSS is particularly important as it measures the improvement of the forecast over the standard forecast that may be due to random chance. The range of HSS is from $-\infty$ to 1. When the value of HSS is negative, it means that the chance forecast is better, zero means no skill, and a value of one indicates a perfect forecast. For all forecasted timestamps, we obtain a minimum HSS of 0.623 for t + 150 minutes which indicates that the performance of our model is much better than that due to the chance forecast even for the 150-minutes forecast. This is also supported by the good ETS which measures the skill of the forecast relative to random or chance events. We also notice that the TS/CSI decreases with lead time which means that the fraction of observed events that are correctly predicted by the model decreases with time.

The reason for the decrease in accuracy or increase in error metric with lead time is due to the fact that in the forecasting of the next consecutive timestamp, we consider the previously forecasted precipitation data. Since the previously predicted precipitation already has some error, the error is propagated during the prediction of the next precipitation data which further gets accumulated in the consecutive predictions eventually leading to large decrease in accuracy or increase in error metric for longer lead time.

The scatter plot of the observed and predicted precipitation, as shown in Fig 8 has high density up to 8 mm/hour as very heavy precipitation (> 8 mm/hour) is not observed very often. We also find that precipitation above 20 mm/hour has a very less density as such precipitation values are rare and are not present in the test dataset. The density of the points is maximum near zero as many cells do not have precipitation in the observed and predicted data. From the scatter plot, we find that the model makes significant errors in predicting precipitation beyond 20 mm/hour. It always underestimates the values of precipitation above 20 mm/hour as we rarely find predicted precipitation above 20 mm/hour. This is also shown by the Multiplicative Bias score, which decreases below one quickly with lead forecast time. The reason for this is the number of training samples is much less for higher precipitation values and because of which Convcast is more biased towards the prediction of lower precipitation values. This is also a general problem with imbalanced dataset in deep learning-based techniques [33]. Convcast, however, estimates the speed and direction of the storms accurately from previous precipitation data and the shape of the forecasted precipitation well corresponds with the observed precipitation. This is because Convcast has learned the spatial correlations between different timestamps from the previous sequences during end-to-end training.

## Conclusion

In this paper, we present a novel architecture Convcast for nowcasting precipitation from spaceborne satellite data. We tune Convcast using a random search on relevant set of hyperparameters in an AutoML toolkit. Results from the hyperparameter tuning suggest that a larger kernel size is better for the spatiotemporal nowcasting problem because of a larger receptive field. Further, we find that simple LSTM models are not suitable for prediction from spatiotemporal data and even a ConvLSTM model with a small receptive field and hidden units outperforms simple LSTM models. We find that our model Convcast nowcasts precipitation in various events such as tropical depression, hurricanes, stratiform system, convective system, etc. in the test dataset with good accuracy even for a lead time of 150 minutes and consistently outperforms state-of-the-art optical flow based methods.

We conclude that Convcast is very suitable for capturing spatiotemporal relations in the satellite-based precipitation dataset for short-term forecasting. The model well preserves the speed and directions of the precipitation in the forecasted results. Satellite-based precipitation nowcasting is quite important as radar data has limitations of not being available in all regions. This is also helpful for future on-board processing of precipitation data.

We find that Convcast, however, is not suitable for nowcasting high precipitation because of its nature to underestimate high precipitation values because of less samples of such precipitation examples in the training data. In our subsequent studies, we would like to work on this problem by increasing the high precipitation training examples, considering the type of precipitation and location information before nowcasting, and using a weighted loss function such as Focal Loss [34]. Significant improvement in the results could be expected using increased training set, pre-classification of storm type with geographical information, and using a weighted loss function.

## Acknowledgments

This research was carried out at the Jet Propulsion Laboratory, California Institute of Technology, under a contract with the National Aeronautics and Space Administration (NASA).

## Author Contributions

**Conceptualization:** Tanvir Islam, Chris Mattmann, Brian Wilson.

**Data curation:** Ashutosh Kumar.

**Formal analysis:** Ashutosh Kumar.

**Funding acquisition:** Tanvir Islam.

**Investigation:** Ashutosh Kumar, Tanvir Islam, Yoshihide Sekimoto, Chris Mattmann, Brian Wilson.

**Methodology:** Ashutosh Kumar, Tanvir Islam.

**Project administration:** Tanvir Islam, Chris Mattmann, Brian Wilson.

**Resources:** Ashutosh Kumar.

**Software:** Ashutosh Kumar.

**Supervision:** Tanvir Islam, Yoshihide Sekimoto, Chris Mattmann, Brian Wilson.

**Validation:** Ashutosh Kumar, Tanvir Islam, Yoshihide Sekimoto, Chris Mattmann, Brian Wilson.

**Visualization:** Ashutosh Kumar, Yoshihide Sekimoto.

**Writing – original draft:** Ashutosh Kumar.

**Writing – review & editing:** Ashutosh Kumar, Tanvir Islam, Yoshihide Sekimoto, Chris Mattmann, Brian Wilson.

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
