## [Decision Letter · Decision Letter 0]

12 Dec 2019

PONE-D-19-30759

Convcast: An embedded convolutional LSTM based architecture for precipitation nowcasting using satellite data

PLOS ONE

Dear Mr. Kumar,

Thank you for submitting your manuscript to PLOS ONE. After careful consideration, we feel that it has merit but does not fully meet PLOS ONE’s publication criteria as it currently stands. Therefore, we invite you to submit a revised version of the manuscript that addresses the points raised during the review process.

As you will see from the reviewers comments, we ask you to please be more experimentally convincing in answering the questions as to why and how Convcast outperforms the state-of-the-art ConvLSTM. Also, please clarify the components of the model and their interaction.

We would appreciate receiving your revised manuscript by Jan 26 2020 11:59PM. To enhance the reproducibility of your results, we recommend that if applicable you deposit your laboratory protocols in protocols.io, where a protocol can be assigned its own identifier (DOI) such that it can be cited independently in the future. For instructions see: http://journals.plos.org/plosone/s/submission-guidelines#loc-laboratory-protocols

We look forward to receiving your revised manuscript.

Kind regards,

Ruxandra Stoean

Academic Editor

PLOS ONE

Journal Requirements:

1. We note that [Figure(s) 1,2,7] in your submission contain [map/satellite] images which may be copyrighted. All PLOS content is published under the Creative Commons Attribution License (CC BY 4.0), which means that the manuscript, images, and Supporting Information files will be freely available online, and any third party is permitted to access, download, copy, distribute, and use these materials in any way, even commercially, with proper attribution. For these reasons, we cannot publish previously copyrighted maps or satellite images created using proprietary data, such as Google software (Google Maps, Street View, and Earth). For more information, see our copyright guidelines: http://journals.plos.org/plosone/s/licenses-and-copyright.

1.    You may seek permission from the original copyright holder of Figure(s) [1,2,7] to publish the content specifically under the CC BY 4.0 license. 

Reviewers' comments:

Reviewer's Responses to Questions

**Comments to the Author**

1. Is the manuscript technically sound, and do the data support the conclusions?

Reviewer #1: Yes

Reviewer #2: Yes

2. Has the statistical analysis been performed appropriately and rigorously? 

Reviewer #1: Yes

Reviewer #2: Yes

3. Have the authors made all data underlying the findings in their manuscript fully available?

Reviewer #1: Yes

Reviewer #2: Yes

4. Is the manuscript presented in an intelligible fashion and written in standard English?

Reviewer #1: Yes

Reviewer #2: Yes

5. Review Comments to the Author

Reviewer #1: I find the results of this paper to be technically sound. The presented metrics demonstrate that proposed algorithm, based on the stack of ConvLSTM layers, works better than the chosen optical flow baselines. The explicit approach to the hyperparameter search, which is often hidden and obscured is an advantage of this work.

However, we already know that ConvLSTM beats optical flow on precipitation nowcasting task from the original paper on ConvLSTM by Xingjian Shi which is cited in the literature review. The only comparison with pure ConvLSTM approach presented in table 6 fails to isolate the reason why Convcast outperforms ConvLSTM. Are three layers better than one? Maybe the 3D convolution is crucial? Or is it because of hyperparameters, and the ConvLSTM baseline is simply crippled because of small 2x2 filters (original paper uses 3x3 and 5x5)? It is unfortunate these questions can't be answered based on the provided experiments

I was confused by the description of dichotomous method (usually called binary classification in ML-related texts). I am used to neural networks trained for classification tasks with softmax or sigmoid activation functions attached to the last layer. In this familiar setting, the output is almost always greater than zero and binarization by comparison with zero is not appropriate, some other threshold have to be chosen. As far as I understand in case of Convcast, ReLU was chosen as the activation function on the last layer, so negative activations turn into zeros (no rain prediction) and it all makes sense both in terms of resulting classification metrics and RMSE. I believe this point should be clarified in the text. Additionally, gradient is not backpropagated through the true negatives because of ReLU, so it may be the natural aid with the imbalance problem.

The notation at line 110 is confusing, use \\times{} instead of X

Reviewer #2: The paper proposed a new architecture for precipitation nowcasting called Convcast. Rather than using the radar echo data, Convcast utilizes the satellite data for precipitation nowcasting. Satellite data are able to reflect rainfall at a higher altitude and are available for the entire globe. Thus, studying the precipitation nowcasting problem with the help of satellite data is a valuable research topic. The Convcast model is based on Convolutional LSTM (ConvLSTM). The author proposed to use automatic machine learning (AutoML) to search for the network structure, which is a nice approach to automate the trial-and-error phase in designing deep learning models. The author illustrates that Convcast achieves state-of-the-art performance by the experiments.

The paper is well written and is a nice addition to the recent trend of incorporating machine learning to solve environmental problems.

6. PLOS authors have the option to publish the peer review history of their article (what does this mean?). If published, this will include your full peer review and any attached files.

Reviewer #1: Yes: Vadim Lebedev

Reviewer #2: No

---

## [Author Response · Author response to Decision Letter 0]

16 Feb 2020

To the academic editor:

We would like to clarify that our intention is to come up with an architecture for precipitation nowcasting from satellite data instead of using Ground-based radar data which most of the previous researches have focused on; as also commented by the reviewer #2. In this sense, we do not wish to outperform the ConvLSTM. In fact, our Convcast architecture consists of several layers of ConvLSTM. ConvLSTM based architecture has been shown to be successful in precipitation nowcasting using Ground-based radar data which motivated us to develop an architecture from ConvLSTM cells for nowcasting using satellite data. However, finding a network architecture for a new problem always requires a lot of trial and errors because of several hyperparameters involved and this is where we make use of the novel Automated Machine Learning (AutoML) to obtain the Convcast architecture which outperforms several optical flow baseline methods on the test dataset as shown experimentally in the paper. Use of AutoML is a novel study area and this research is one of the very first ones that incorporates AutoML for precipitation nowcasting from satellite data. We have now included another ConvLSTM based architecture (please see response to the reviewer #1) for more experimentally convincing the superiority of Convcast and the effect of kernel size and filters on the validation loss. We also have now better clarified the components of the model and their interaction.

After talking to co-authors, I came to know that no map file is used. Rather, the plots are generated by us, and coastal lines are simply overlaid. No licenses are involved here. All map files and images are produced by us and coming directly from NASA’s IMERG data and they have already been projected to grids. There is no involvement of any kind of licenses and all plots are produced by us.

The dataset used in the visualization has now been cited in the manuscript in accordance with data citation format mentioned on the NASA Precipitation Measurement Missions website https://pmm.nasa.gov/data-access/citations. The reference number is [28] in the revised manuscript.

To the reviewer #1:

We wanted to show the extreme cases and that is why we compared 2 x 2 ConvLSTM with the Convcast which has 7 x 7. In the discussion, we wrote that the accuracy of the model increases with the increase in kernel size because a larger kernel has a larger receptive field. We have now included a 5 x 5 baseline in Table 6 to show the effect of larger kernel on decreasing loss. 3D Convolution is present in all the ConvLSTM networks which has been used to get precipitations from the last ConvLSTM layer’s activations before it passes through ReLU to suppress any negative values to zero. More deeper networks are better than shallow networks. However, in our research all ConvLSTM baselines have the same backbone as the Convcast. Thus, it is because of the right choice of the hyperparameters obtained from the AutoML experiment which makes the Convcast outperform other ConvLSTM baseline methods. We have made the required changes in the manuscript.

This is a common terminology in weather forecast/nowcast. We have provided the proper definitions of dichotomous method (binary classification) and would like to keep it the same way. Most of the papers on forecasting/nowcasting use the same terminology and we would like to maintain the consistency. It has now been clarified in the text.

Yes, in classification related tasks with softmax or sigmoid activation functions, the output value (probability) will be greater than zero in most of the cases and binarization is made using a threshold greater than zero (generally 0.5). 

As mentioned by you, the output of our last layer is not probability rather precipitation value, and thus we have chosen ReLU in the last layer. In fact, an activation function in the last layer might not be necessary (we can use y = x) as it is totally a regression problem. However, since we know from our experience that precipitation cannot take negative values, we put ReLU as an activation function at the last layer to turn negative activations to zero. Again, the output of ReLU can be any value greater than or equal to zero. We have now clarified this point in the text.

Thank you for pointing it out. We have replaced X with \\times{}

To the reviewer #2:

Thank you for your comments!

---

## [Editor Report · Decision Letter 1]

24 Feb 2020

Convcast: An embedded convolutional LSTM based architecture for precipitation nowcasting using satellite data

PONE-D-19-30759R1

Dear Dr. Kumar,

We are pleased to inform you that your manuscript has been judged scientifically suitable for publication and will be formally accepted for publication once it complies with all outstanding technical requirements.

With kind regards,

Ruxandra Stoean

Academic Editor

PLOS ONE
---

## [Editor Report · Acceptance letter]

28 Feb 2020

PONE-D-19-30759R1 

Convcast: An embedded convolutional LSTM based architecture for precipitation nowcasting using satellite data 

Dear Dr. Kumar:

I am pleased to inform you that your manuscript has been deemed suitable for publication in PLOS ONE. Congratulations! Your manuscript is now with our production department. 

With kind regards,

on behalf of

Dr. Ruxandra Stoean 

Academic Editor

PLOS ONE